# Bell's theorem for temporal order

Magdalena Zych [1], Fabio Costa [1], Igor Pikovski [2,3,4] & Časlav Brukner[5,6]

Time has a fundamentally different character in quantum mechanics and in general relativity. In quantum theory events unfold in a fixed order while in general relativity temporal order is influenced by the distribution of matter. When matter requires a quantum description, temporal order is expected to become non-classical—a scenario beyond the scope of current theories. Here we provide a direct description of such a scenario. We consider a thought experiment with a massive body in a spatial superposition and show how it leads to entanglement of temporal orders between time-like events. This entanglement enables accomplishing a task, violation of a Bell inequality, that is impossible under local classical temporal order; it means that temporal order cannot be described by any pre-defined local variables. A classical notion of a causal structure is therefore untenable in any framework compatible with the basic principles of quantum mechanics and classical general relativity.

[1] Centre for Engineered Quantum Systems, School of Mathematics and Physics, The University of Queensland, St. Lucia, QLD 4072, Australia. [2] ITAMP, Harvard-Smithsonian Center for Astrophysics, Cambridge, MA 02138, USA. [3] Department of Physics, Harvard University, Cambridge, MA 02138, USA. [4] Stevens Institute of Technology, Hoboken, NJ 07030, USA. [5] Vienna Center for Quantum Science and Technology (VCQ), Faculty of Physics, University of Vienna, Boltzmanngasse 5, 1090 Vienna, Austria. [6] Institute for Quantum Optics and Quantum Information (IQOQI), Austrian Academy of Sciences, Boltzmanngasse 3, 1090 Vienna, Austria. Correspondence and requests for materials should be addressed to M.Z. (email: m.zych@uq.edu.au)

Quantum mechanics forces us to question the view that physical quantities (such as spin, positions or energy) have predefined values: Bell's theorem shows that if observable quantities were determined by some locally defined classical variables, it would be impossible to accomplish certain tasks—such as the violation of Bell's inequalities—whereas such tasks are possible according to quantum mechanics[1,2] and have been realised in experiments[3–6]. However, the causal relations between events remain fixed in quantum theory: whether an event A is in the past, in the future, or space-like separated from another event B is predefined by the location of such events in space-time[7,8]. In contrast, in general relativity, space-time itself is dynamical: the presence of massive objects affects local clocks and thus causal relations between events defined with respect to them. Nonetheless, the dynamical causal structure of general relativity is still classically predefined: the causal relation between any pair of events is uniquely determined by the distribution of matter-energy degrees of freedom (DOFs) in their past light cone. In other words, causal relations are always determined by local classical variables. This picture is expected to change if we consider quantum states of gravitating DOFs: if a massive system is prepared in a superposition of two distinct states, each yielding an observably different causal structure for future events, would it be possible to observe causal relations that display genuine quantum features?

A main obstacle in the analysis of macroscopic superpositions of gravitating bodies is that, in the absence of a classical space-time manifold, it becomes unclear how to identify space-like surfaces on which quantum states are defined, or global fields of time-like vectors to define time evolution. Indeed, some models even postulate that such superpositions are simply not valid physical states and must decohere (or collapse) fast enough to preserve a classical description of space-time and dynamical laws[9–13]. A very different mindset underlies various quantum gravity frameworks[14]—where quantum features of the metric and therefore of the causal relations are indeed expected. However, to date, none of the quantum gravity frameworks has been applied to analyse such an epitomic example as superpositions of space-times with macroscopically distinct causal structures. Therefore, it is unclear whether there exists any phenomenology unequivocally associated with quantum causal structures, nor whether quantum gravity frameworks can circumvent or directly address the objections against superpositions of manifolds. Independently, quantum formalisms have been recently developed to study quantum causal structures at an abstract level in the context of quantum-information processing[8,15,16]. However, although quantum features of space-time are among the motivations for these studies, no direct link with quantum gravity has yet been established.

This work provides the first direct analysis of quantum causal relations arising from a spatial superposition of a massive object. We show how the temporal order between time-like events can become superposed or even entangled. We further discuss a thought experiment, an admissible albeit remote physical scenario, where these non-classical causal relations arise among physical events. In order to prove their non-classicality, we formulate a Bell-type theorem for temporal order: We define a task that cannot be accomplished if the time order between the events was predetermined by local variables, while the task becomes possible if the events are in a space-time region affected by the gravitational field of a massive object in an appropriate quantum state. Our approach provides a method to directly describe scenarios so far considered to be out of reach for standard theoretical physics. We show explicitly how to overcome the difficulties with describing superpositions of metrics that motivated collapse models. On the other hand, our result is independent of the high-energy completion of any specific quantum gravity framework—we do not assume any new physics, the results are based entirely on well-established, low-energy general relativity and on quantum mechanics. Our results are therefore robust against particular mathematical approaches to quantising gravity, thus providing a benchmark for specific frameworks. Furthermore, the time and energy scale at which entangled temporal order arises is closer than the Planck scale, typically invoked in this context, and is also far remote from the scale given by the decoherence models—which therefore do not preclude quantum features of space-time to arise. Our results thus reveal that both the above approaches are missing crucial intuition and correct physical understanding of the phenomena associated with causal structures at the interface of quantum and gravitational physics. In turn, our work provides a robust method to quantitatively assess these phenomena, helping to build correct physical intuition for quantum causal structures.

## Results

**Dynamical causal structure in general relativity.** In classical general relativity, the causal structure is the structure of light cones of the space-time metric[17,18]. As the matter-energy DOFs determine the metric through Einstein's equations, the causal structure of a region of space-time is dynamical: it depends on the state of the matter energy in its past light cone. A major obstacle towards a quantum theory of gravity is that it is not clear how to transpose the mathematical notion of causal relations to scenarios where matter DOFs can be in general quantum states, as such scenarios seem to preclude the use of any underlying space-time manifold with respect to which events, light cones and causal relations could be defined. To overcome this obstacle, our approach is to start from a physical understanding of events and their causal relations. Even in classical general relativity a physical event cannot be directly identified with a point on a space-time manifold, a fundamental aspect of the theory captured mathematically by diffeomorphism invariance[19]. Although it can be debated whether or not space-time points have an intrinsic physical meaning, a natural way to define diffeomorphism-invariant events is to specify them operationally, relative to physical systems; for example, positions and proper times of physical systems used as clocks[20]. We adopt this notion of events throughout the work. Causal relations are then understood as the possibility to exchange non-faster-than-light signals—or more generally, physical systems—between operationally defined events.

The presence of massive bodies generally alters the relative rates at which clocks tick. For example, in a weak field limit, a clock in a gravitational potential $\Phi$ exchanging signals with an identical clock far away from the source of $\Phi$, where the potential effectively vanishes, will appear to tick slower by a factor $\sqrt{1 + 2\frac{\Phi}{c^2}}$. In classical physics, this leads to the well-tested time-dilation[21,22] and redshift effects[23]. When the clocks are described as quantum systems, new effects arise from the combination of quantum and general relativistic theories. For a clock in superposition of different distances to the mass, its time-keeping DOFs become entangled to the clock's position[24–26]. This entanglement implies a universal decoherence mechanism for generic macroscopic systems under time dilation[27,28]. The regime of low-energy quantum systems in curved space-time can be described within a framework of general relativistic composite quantum particles[29]. Here we additionally exploit the fact that only the distance between a clock and a mass has physical significance and due to linearity of quantum theory this must hold also for a superposition of different distances. (There is no difference in the relative ticking rates of two clocks whether we

think that the clocks are being positioned at different distances—possibly in a superposition—from the mass, or that the mass is positioned at different distances from the clocks[30].)

Consider two agents, a and b, with two initially synchronised clocks, each following a fixed world line. A third agent prepares one of two mass configurations, $K_{A \prec B}$ or $K_{B \prec A}$, so as to induce time dilation between the clocks of a and b. If configuration $K_{A \prec B}$ is prepared, event A—defined by the clock of agent a showing proper time $t_a = \tau^*$—will be in the past light cone of the event B, which is defined in an analogous way: by the clock of agent b showing proper time $t_b = \tau^*$. If configuration $K_{B \prec A}$ is prepared, event B will be in the past light cone of event A. To keep the world lines of the agents independent of the mass configuration, their laboratories can be embedded in tight enough trapping potentials, that is, much stronger than the gravitational field (which is feasible since our protocol does not require macroscopic source masses, see Methods). In Supplementary Note 4 we discuss other mass configurations, which have the desired effect on temporal order, but for which the agents a, b can remain inertial.

A possible way to realise configuration $K_{A \prec B}$ is to place an approximately point-like body of mass $M$ closer to b than to a, see Fig. 1. The light-cone structure of the resulting space-time is fully determined by the metric tensor $g_{\mu\nu}$, for which we adopt the sign convention $(-, +, +, +)$. In isotropic coordinates in the first-order post-Newtonian expansion the metric components are[31] $g_{00}(r) = -(1 + 2\frac{\Phi(r)}{c^2})$ and $g_{ij}(r) = \delta_{ij}(1 + 2\frac{\Phi(r)}{c^2})^{-1}$, $i, j = 1, 2, 3$, where $\Phi(r) = -\frac{GM}{r}$ is the gravitational potential and $r$ is the spatial distance between the mass and the event where the metric is evaluated. For an event with a spatial coordinate $\mathbf{R}_a$ and the mass at a spatial coordinate $\mathbf{r}_M$ (where the spatial coordinates are defined, for example, by a far-away agent as in Fig. 1), we have $r \equiv |\mathbf{R}_a - \mathbf{r}_M|$. Note that we use a common coordinate system to describe the different mass configurations and the associated space-time metrics. Operationally, we can associate such coordinates with the far-away agent, whose local clocks are not affected by the change in the matter distribution. However, this is only a convenient interpretation, we can always think of the coordinates in analogy to gauge fixing—any physical prediction regarding proper times of the clocks and exchange of the signals will not depend on the choice of coordinates.

We consider that a and b remain at fixed coordinate distances from the mass, $r_a$ and $r_b = r_a - h$, respectively, and find the parameters for which event A ends up in the past light cone of B for $K_{A \prec B}$ (and vice versa for $K_{B \prec A}$). An infinitesimal proper time element along a world line at a distance $r$ from the mass is given by $d\tau(r) = \sqrt{-g_{00}(r)}dt$, where $t$ is the coordinate time, and a photon travelling in the radial direction from $r_a$ reaches $r_b$ after a coordinate time $T_c = \frac{1}{c}\int_{r_b}^{r_a} dr' \sqrt{-\frac{g_{rr}(r')}{g_{00}(r')}}$. Therefore, if the photon is emitted at the local time $t_a = \tau^*$, it reaches $r_b$ when b's local time is $\bar{t}_b = \sqrt{-g_{00}(r_b)}\left(\frac{\tau^*}{\sqrt{-g_{00}(r_a)}} + T_c\right)$, assuming that the local clocks are synchronised so that $t_a = 0$ and $t_b = 0$ coincide with the coordinate time $t = 0$. For

$$\tau^* > T_c \frac{\sqrt{-g_{00}(r_b)}}{1 - \sqrt{\frac{g_{00}(r_b)}{g_{00}(r_a)}}}, \qquad (1)$$

we have $\bar{t}_b \leq \tau^*$, which means that there is enough time for a not-faster-than-light signal emitted at event A (defined by $t_a = \tau^*$) to travel the distance $h$ and reach agent b at event B (defined by $t_b = \tau^*$). This means that event A is in the causal past of event B as required. For example, for $h \ll r_a$ condition (1) is satisfied for $\tau^* > \frac{2r_a^2 c}{GM}$. Configuration $K_{B \prec A}$ can be arranged analogously, by

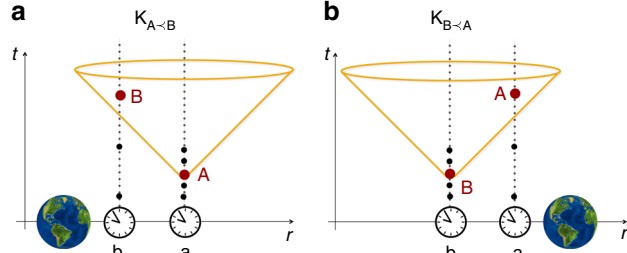

**Fig. 1** General relativistic engineering of causal relations between space-time events using a massive body. Initially synchronised clocks a and b are positioned at fixed distances from a far-away agent whose time coordinate is $t$. Event A (B) is defined by the clock of a (b) showing proper time $\tau^*$. In configuration $K_{A \prec B}$ (left) a mass is placed closer to b than to a. Due to gravitational time dilation, event A can end up in the causal past of event B: for a sufficiently large $\tau^*$ the time difference between the clocks becomes greater than it takes light to travel between them. Light emitted at event A reaches clock b before the event B occurs. Configuration $K_{B \prec A}$ (right) is fully analogous to $K_{A \prec B}$: the mass is placed closer to clock a and the event B can end up in the causal past of the event A

placing the mass closer to a than to b. Then, the condition $\tau^* > \frac{2r_b^2 c}{GM}$, for $h \ll r_b$, ensures that B is in the causal past of A. Note that with the above conditions on $\tau^*$ the events A and B are always time-like separated, but have different time orders for the two mass configurations—these conditions guarantee that the time order between A and B is swapped in all reference frames.

The example above simply illustrates that in general relativity causal structure is dynamical and depends on the stress-energy tensor of the matter DOFs: preparing different matter distributions on a space-like hypersurface can result in different causal relations between events in its causal future.

**Quantum control of temporal order**. When A is in the past light cone of B, a physical system can in principle be transferred from A to B. Consider a quantum system S initially prepared in state $|\psi\rangle^S$, which undergoes a unitary $U_A$ at event A (at the space-time location where the clock of agent a marks proper time $\tau^*$) and a unitary $U_B$ at event B. Such ordered events can therefore result in the following state of S:

$$\left|\tilde{\psi}_1\right\rangle^S = U_B U_A |\psi\rangle^S. \qquad (2)$$

If B is before A, and S is prepared in the same initial state, the final state of S is

$$\left|\tilde{\psi}_2\right\rangle^S = U_A U_B |\psi\rangle^S. \qquad (3)$$

A situation can therefore be arranged such that state (2) is produced for configuration $K_{A \prec B}$ and (3) is produced for $K_{B \prec A}$. (We ignore a possible additional time evolution between the two events for simplicity.) Different mass configurations can result in different temporal orders of local operations, which holds in quantum as well as in classical theory. Let us make the following assumptions:

(a) Macroscopically distinguishable states of physical systems can be assigned orthogonal quantum states.
(b) Gravitational time dilation in a classical limit reduces to that predicted by general relativity.
(c) The quantum superposition principle holds (regardless of the mass or nature of the involved system).

Even though the above assumptions hold in the standard quantum and general relativistic frameworks, it is not known if a fundamental theory of quantum gravity satisfies them. Our aim is

to investigate their consequences for the notion of temporal order.

The coordinates introduced in the previous section define a foliation of space-time into equal-time slices. As long as no horizons are present in any of the considered configurations, such slices define space-like hypersurfaces. With each hypersurface one can associate a Hilbert space, containing the quantum states of interest at the given time. The time coordinate corresponds to the time $t$ in Fig. 1 and is operationally defined as the time measured by the local clock of the far-away agent (not affected by the mass configurations). These quantum states can be understood operationally as states assigned by the far-away agent. However, as discussed in the previous section, such an interpretation is not strictly necessary but is merely a convenient way to define the relevant mathematical objects and to carry out the calculations.

The two mass configurations $K_{A \prec B}$, $K_{B \prec A}$ can thus be assigned quantum states $\left|K_{A \prec B}\right\rangle^M$, $\left|K_{B \prec A}\right\rangle^M$. By assumption (a) these states are orthogonal. Since each state individually satisfies the classical limit (mass is sufficiently localised around a single world line), following assumption (b), the system S will evolve as in Eqs. (2) or (3) depending whether the mass is in state $\left|K_{A \prec B}\right\rangle^M$ or $\left|K_{B \prec A}\right\rangle^M$, respectively. Finally, by assumption (c), a superposition $\left|K_+\right\rangle^M := \frac{1}{\sqrt{2}}\left(\left|K_{A \prec B}\right\rangle^M + \left|K_{B \prec A}\right\rangle^M\right)$ is a physically allowed mass configuration, and will yield the following final state of the joint system:

$$\left|\psi_{\text{sup}}\right\rangle^{MS} = \frac{1}{\sqrt{2}}\left(\left|K_{A \prec B}\right\rangle^M U_B U_A \left|\psi\right\rangle^S + \left|K_{B \prec A}\right\rangle^M U_A U_B \left|\psi\right\rangle^S\right). \tag{4}$$

An explicit calculation showing how this state arises is presented in Methods. We note that not only classical gravity but also semi-classical[14] and stochastic gravity[32] theories would not yield Eq. (4) since these frameworks describe gravitational interactions in terms of classical, possibly stochastic, variables, thus violating assumption (c).

Note that, given a specific physical system used as a clock, it is possible to simulate its time dilation using non-gravitational interactions. For example, an electric field can shift atomic energy levels and thus "time dilate" a clock based on atomic transitions. Therefore, one can produce a state analogous to (4) without using gravity. However, only gravity can alter the relative ordering of events independently of the nature of the systems and interactions used as clocks, due to the universality of time dilation: the preparation and manipulation of the massive object can be carried out without any knowledge of other aspects of the protocol. Such a universality underpins a fundamental distinction between our gravitational protocol and other, non-gravitational, methods to control causal relations between operationally defined events[33–39]. (See also Supplementary Note 4 for further discussion.)

Finally, the state (4) is the result of a process wherein the order of operations on a target system (S) is determined by the quantum state of a control system (position of the massive body). Such a process is known as a quantum switch[15] and has been studied as a possible quantum-information resource[40–44]. The state $\left|\psi_{\text{sup}}\right\rangle^{MS}$ is a superposition of two amplitudes corresponding to different predefined, classical orders between events A and B. Note that, if the control system is discarded, the reduced state of S is

$$\frac{1}{2}\left(\left|\tilde{\psi}_1\right\rangle\left\langle\tilde{\psi}_1\right|^S + \left|\tilde{\psi}_2\right\rangle\left\langle\tilde{\psi}_2\right|^S\right), \tag{5}$$

which is indistinguishable from a probabilistic mixture of $\left|\tilde{\psi}_1\right\rangle$ and $\left|\tilde{\psi}_2\right\rangle$. The state in Eq. (5) can be interpreted as arising from

events A and B with a classical, albeit unknown, temporal order. Therefore, any protocol aimed at testing operationally quantum features of temporal order necessarily requires a measurement of the control system.

**Bell's theorem for temporal order**. The above argument shows that superpositions of massive objects can in principle result in a coherent quantum control of temporal order between events. However, one might question whether such a conclusion has a direct physical meaning or whether it relies on a particular interpretation of state (4). Furthermore, the state assignment is defined in terms of a given coordinate system, while we would like to base our conclusions on coordinate-independent physical events. Since the very meaning of quantum states and measurements might be put into question in the absence of a classical space-time, a proof of non-classical causal relations should not rely on the validity of the quantum formalism. In the following we show that it is possible to probe the nature of temporal order irrespective of the validity of quantum theory. We formulate a theory-independent argument—which does not rely on the quantum framework and provides means to exclude the very possibility of explaining data from a hypothetical experiment in terms of a classical temporal order (which can be stochastic and dynamical) within a broad class of probabilistic theories, not limited to quantum mechanics. Our formulation is analogous to Bell's theorem for local hidden variables[1,2] (see Methods) and we thus refer to the theorem below as Bell's theorem for temporal order of events. The core of the argument is simple: given a bipartite system prepared in a separable state, it is not possible to violate any bipartite Bell inequality by performing local operations (transformations and measurements) on the two parts, as long as the local operations are applied in a definite order.

The scenario involves a bipartite system with subsystems $S_1$ and $S_2$ and a system M that can influence the temporal order of events. For $j = 1, 2$, each system $S_j$ undergoes two transformations, $T_{A_j}$ and $T_{B_j}$, at space-time events $A_j$, $B_j$, respectively. Each system is then measured at an event $C_j$ according to some measurement setting $i_j$, producing a measurement outcome $o_j$. Additionally, M is measured at an event D, space-like separated from both $C_1$ and $C_2$, producing an outcome $z$, see Fig. 2. We now define the notion of classical order between events:

**Definition 1**: A set of events is classically ordered if, for each pair of events A and B, there exists a space-like surface and a classical variable $\lambda$ defined on it that determines the causal relation between A and B: for each given $\lambda$, either $A \preceq B$ (A in the past causal cone of B), $B \preceq A$ (A in the past causal cone of B) or $A \| B$ (A and B space-like separated).

Classically ordered events do not necessarily form a partially ordered set: classical order can be dynamical (the order between two events can depend on some operation performed in the past, i.e. some agent can prepare $\lambda$) and stochastic ($\lambda$ might be distributed according to some probability, and not specified deterministically)[45,46].

*Bell's theorem for temporal order*. No states, set of transformations and measurements which obey assumptions 1–5 below can result in a violation of the Bell inequalities:

1. Local state: The initial state $\omega$ of $S_1$, $S_2$ and M is separable (as defined in Methods).
2. Local operations: All transformations performed on the systems are local (as defined in Methods).
3. Classical order: The events at which operations (transformations and measurements) are performed are classically ordered.

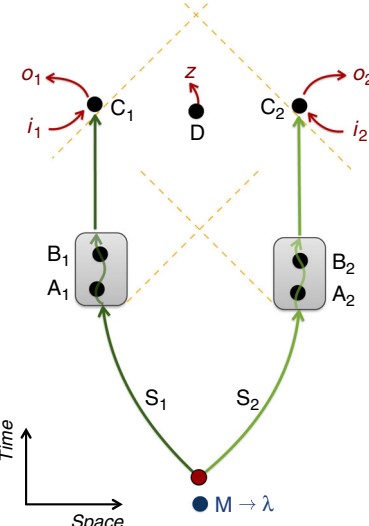

**Fig. 2** Bell's theorem for temporal order. A bipartite system, made of subsystems $S_1$ and $S_2$, is sent to two groups of agents. Operations on $S_1$ ($S_2$) are performed at events $A_1$, $B_1$ ($A_2$, $B_2$). At event $C_1$ ($C_2$), a measurement with setting $i_1$ ($i_2$) and outcome $o_1$ ($o_2$) is performed. Events $A_1$, $B_1$ are space-like separated from $A_2$, $B_2$ and $C_1$ is space-like to $C_2$; light cones are marked by dashed yellow lines. The order of events $A_j$, $B_j$, $j = 1$, 2, is described by a variable $\lambda$ defined by a system M. The system M is measured at event D, producing an output bit $z$. If the initial state of the systems $S_1$, $S_2$, M is separable, and $\lambda$ is a classical variable (possibly dynamical and probabilistic), the resulting bipartite statistics of the outcomes $o_1$, $o_2$ cannot violate any Bell inequality, even if conditioned on $z$

4. Space-like separation: Events ($A_1$, $B_1$) are space-like separated from events ($A_2$, $B_2$); $C_1$, $C_2$, and D are pair-wise space-like separated.
5. Free-choice: The measurement choices in the Bell measurement are independent of the rest of the experiment. (This is a standard assumption necessary in Bell-like theorems.)

More formally, let us denote by $\mathbb{T} = (T_{A_1}, T_{B_1}, T_{A_2}, T_{B_2})$ the set of all local transformations irrespective of their order. The thesis of the theorem can be rephrased as: the conditional probability

$$P(o_1, o_2 | i_1, i_2, z, \mathbb{T}, \omega) \qquad (6)$$

produced under assumptions 1–5 does not violate Bell's inequalities for any value of $z$. The proof of the theorem is presented in Methods.

**Violation of Bell inequalities for temporal order**. Here we show how the gravitational quantum control of temporal order from the first section can result in events whose temporal order is entangled: a bipartite quantum system, initially in a product state $|\psi_1\rangle^{S_1} |\psi_2\rangle^{S_2}$, is sent to two different regions of space such that $a_1$, $b_1$ and $c_1$ only interact with $S_1$, while $a_2$, $b_2$ and $c_2$ only interact with $S_2$. Agents $a_1$, $a_2$ perform, respectively, the unitaries $U_{A_1}$, $U_{A_2}$ at the events $A_1$, $A_2$, while agents $b_1$, $b_2$ perform the unitaries $U_{B_1}$, $U_{B_2}$ at the events $B_1$, $B_2$. Finally, $c_1$ and $c_2$ measure $S_1$ and $S_2$ at events $C_1$ and $C_2$, respectively, see Fig. 3. Assume that a massive system can be prepared in two configurations, $K_{A \prec B}$ and $K_{B \prec A}$, such that $A_1 \prec B_1 \prec C_1$ ($A_1$ in the past light cone of $B_1$, etc.) and $A_2 \prec B_2 \prec C_2$ for $K_{A \prec B}$, while $B_1 \prec A_1 \prec C_1$ and $B_2 \prec A_2 \prec C_2$ for $K_{B \prec A}$, and such that the events are space-like separated as per assumption 4, which can always be achieved by having the groups sufficiently separated. If the mass is prepared in superposition $\frac{1}{\sqrt{2}} \left( |K_{A \prec B}\rangle^M + |K_{B \prec A}\rangle^M \right)$, the joint state of the

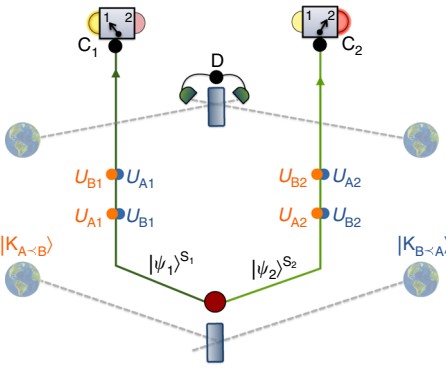

**Fig. 3** Schematics of a protocol for a violation of Bell's inequalities for temporal order. Systems $S_1$, $S_2$ are prepared in a product state $|\psi_1\rangle^{S_1} |\psi_2\rangle^{S_2}$ and sent to space-like separated regions. One pair of agents performs unitary operations $U_{A_1}$, $U_{B_1}$ on $S_1$ at the correspondingly marked space-time events; another pair acts on $S_2$ with unitary operations $U_{A_2}$ and $U_{B_2}$. Each operation is applied only once, at an event defined by the specific proper time of the local clock of the agent. A massive body is prepared in a superposition of two configurations $|K_{A \prec B}\rangle$ and $|K_{B \prec A}\rangle$, which define different causal structures for future events. For the amplitude $|K_{A \prec B}\rangle$, the operations $U_{A_i}$, $i = 1$, 2 applied on $S_i$ are in the causal past of the operations $U_{B_i}$ (orange dots); and vice versa for $|K_{B \prec A}\rangle$ (blue dots). The operations can be chosen such that $U_{A_i} U_{B_i} |\psi_i\rangle^{S_i}$ is orthogonal to $U_{B_i} U_{A_i} |\psi_i\rangle^{S_i}$ (for both $i = 1$ and $i = 2$), resulting in a maximally entangled final state. Bell measurements are performed at events $C_1$ and $C_2$ on $S_1$ and $S_2$, respectively. At event D the mass is measured in a superposition basis. Conditioned on the outcome of this measurement, the results of the measurements at $C_1$, $C_2$ can maximally violate Bell's inequalities, which would not be possible if the order of events was classical (even if probabilistic)

mass and the systems after the application of the unitaries is

$$\frac{1}{\sqrt{2}} \left( |K_{A \prec B}\rangle^M U_{B_1} U_{A_1} |\psi_1\rangle^{S_1} U_{B_2} U_{A_2} |\psi_2\rangle^{S_2} + |K_{B \prec A}\rangle^M U_{A_1} U_{B_1} |\psi_1\rangle^{S_1} U_{A_2} U_{B_2} |\psi_2\rangle^{S_2} \right). \qquad (7)$$

Agent d at the event D measures the mass in the superposition basis $|\pm\rangle = \frac{1}{\sqrt{2}} \left( |K_{A \prec B}\rangle \pm |K_{B \prec A}\rangle \right)$. Conditioned on the outcome, the joint state of $S_1$ and $S_2$ reads

$$\frac{1}{\sqrt{2}} \left( U_{B_1} U_{A_1} |\psi_1\rangle^{S_1} U_{B_2} U_{A_2} |\psi_2\rangle^{S_2} \pm U_{A_1} U_{B_1} |\psi_1\rangle^{S_1} U_{A_2} U_{B_2} |\psi_2\rangle^{S_2} \right). \qquad (8)$$

If the states $U_{B_1} U_{A_1} |\psi_1\rangle^{S_1}$, $U_{B_2} U_{A_2} |\psi_2\rangle^{S_2}$ are orthogonal to $U_{A_1} U_{B_1} |\psi_1\rangle^{S_1}$, $U_{A_2} U_{B_2} |\psi_2\rangle^{S_2}$, respectively, then the state (8) is maximally entangled. Local measurements can thus be performed on subsystems $S_1$, $S_2$ whose outcomes will violate Bell inequalities, conditioned on the measurement outcome at D (see Supplementary Note 2 for an example).

The above thought experiment can in principle be realised in a scenario where it is meaningful to argue that assumptions 1, 2 and 4, 5 are satisfied. Violation of the Bell's inequality would then imply that assumption 3 does not hold, proving non-classicality of temporal order. In order to maximally violate the inequality, the time-dilated clocks of the agents need to decorrelate from the systems $S_i$. In the Methods section we present a particular scenario using photons that satisfies also this requirement. In Supplementary Note 3 we present two concrete examples of our thought experiment, using as the systems $S_i$ polarisation states of photons, depicted in Supplementary Fig. 1, or spatial modes of a quantum field, depicted in Supplementary Fig. 2.

## Discussion

The non-classical causal structures discussed in this work arise in a semi-classical, albeit non-perturbative, regime where no explicit quantisation of the gravitational field is needed (which is complementary to the regime of most quantum gravity frameworks[14]). Our approach shows that general relativity and standard quantum mechanics are sufficient to analyse scenarios involving superpositions of macroscopically different classical backgrounds. Not only is there no tension between the two frameworks, but there is also no ambiguity in the prediction of physical effects that arise: for each probability amplitude, the time-dilation effects introduced by the mass can be treated classically. The considered processes involve a simple superposition of such amplitudes and the final probability amplitude is given by the usual Feynman sum. Note that, even though no explicit quantisation of the metric is used, the amplitudes in the Feynman sum do correspond to macroscopically distinct space-time metrics: this is because each of these amplitudes contains a different causal structure, which determines the metric up to a conformal factor[17,18]. Quantisation of the metric is therefore implicit in our result, in a similar way as in recently considered witnesses for quantum gravity in interferometric scenarios[47–49].

A practical realisation of the Bell test for time order would be extremely challenging, even in light of current efforts to prepare superposition states of massive objects and test their gravitational interactions[50–54]. However, there would be far reaching consequences if a such a test were fundamentally impossible: this would imply that time order, and thus time itself, can be described with a classical parameter even in space-times originating from a quantum state of a massive object—with no need to invoke any other mechanism, such as refs. [9–13], that would decohere these states (see also Supplementary Note 5 for further discussion). On the other hand, since these mechanisms postulate a specific decoherence time of spatial superpositions, one could think that they preclude the preparation of non-classical causal structures. This is not the case: the time required to complete our protocol can be shorter than the decoherence time postulated by these models (see Methods). Thus, contrary to some motivations[11,13], these models do not enforce fundamentally classical space-time with a fixed causal structure (i.e. there is a parameter regime where entangled causal structures could form but decoherence postulated by these models is negligible). Finally, classical temporal order could not be excluded also in a scenario where massive bodies can be prepared in quantum states but one (or more) of the assumptions 1, 2, 4, and 5 cannot be satisfied for some fundamental reason. We note that in particular the notion of locality may be fundamentally limited in the context of quantum gravity[55,56].

We should note that proof-of-principle realisations of indefinite causal order, analogous to the examples discussed here, have been realised in the laboratory. However, such realisations cannot be interpreted as proofs of non-classical space-time in the sense of general relativity, see Supplementary Note 4 for a discussion of the key differences between the gravitational and other methods for a quantum control of temporal order. The full extent of the relation between gravitational and non-gravitational realisations of quantum causal structures merits an in-depth study on its own.

A crucial aspect of Bell's theorem for temporal order is that it provides a theory independent result—it applies to any framework where causal relations are described classically, such as classical, semi-classical[14] and stochastic gravity[32] theories. Moreover, joint validity of the quantum superposition principle and gravitational time dilation, assumptions (a)–(c), suffice for a maximal possible violation of the bound. Therefore, a classical notion of temporal order is untenable in any theory compatible with these basic principles. Finally, the way in which a non-classical causal structure can be engineered exploiting time dilation from a massive body in a quantum state reveals a close connection between the information-theoretic framework of quantum combs/process matrices and joint effects of quantum mechanics and general relativity.

## Methods

**Quantum gravitational control of temporal order.** According to the Einstein equations, a massive object gives rise to a space-time metric $g_{\mu\nu}$, $\mu$, $\nu = 0, ..., 3$, which in isotropic coordinates and a post-Newtonian expansion reads[31]:
$g_{00}(r) = -\left(1 + 2\frac{\Phi(r)}{c^2}\right)$, $g_{ij}(r) = \delta_{ij}\left(1 - 2\frac{\Phi(r)}{c^2}\right)$, $i, j = , 2, 3$, where $r$ denotes the distance to the location of the mass. In other words, if a test mass or a clock is positioned at a spatial coordinate $\mathbf{R}_a$ as described by a far-away agent (as in Fig. 1) and the massive object is at a coordinate $\mathbf{r}_M$, then $r = |\mathbf{R}_a - \mathbf{r}_M|$, which for clarity we denote below by $R_a - r_M$. It is important to note that the same coordinates describe scenarios where the mass is placed at different locations at a finite distance from $\mathbf{r}_M$, as long as it remains far from an asymptotic region so that the spatial and temporal coordinates of the far-away agent remain unaffected (i.e. are those of flat Minkowski space-time). In these coordinates, the Hamiltonian of a clock—a particle with internal DOFs—reads

$$H_a = \sqrt{-g_{00}(R_a - r_M)(\Omega_a^2 + c^2 g_{ij}(R_a - r_M)P^i P^j)}, \qquad (9)$$

(see e.g. refs. [57–59]) where $P^i$, $i = 1, 2, 3$ are the components of the momentum operator, and $\Omega_a$ is the internal Hamiltonian, describing the local time evolution of the internal DOFs. Note that we can restrict ourself to an effectively one-dimensional scenario, so only one of the spatial coordinates has been kept in the above expression. In the first post-Newtonian expansion and considering that both the mass and the clock follow fixed world lines at constant $\mathbf{R}_a$ and $\mathbf{r}_M$, respectively, Eq. (9) becomes

$$H_a \approx \Omega_a \left(1 + \frac{\Phi(R_a - r_M)}{c^2}\right). \qquad (10)$$

The asymptotic time coordinate $t$ defines space-like hypersurfaces that are independent of the location of the mass and on which one can define states of all the involved systems (the clocks, the target systems and the mass itself) and Hamiltonian (10) describes their time evolution of with respect to $t$. Due to the interactions between the mass and the clocks—effected by the space-time metric, which contains the potential $\Phi(R_a - r_M)$—the time evolution of the clocks depends on their relative distance $R_a - r_M$ to the mass. Crucially, by the definition of $t$ and the Hamiltonian our description includes both considered different mass configurations: the mass can be semi-classically localised around a single spatial coordinate $r$ or in superposition of different spatial coordinates and the associated states belong to the same Hilbert space associated with a space-like hypersurface labelled by $t$. We thus have all the tools to analyse time evolution in the presence of a superposition state of the mass, even though it leads to a quantifiably non-classical causal structure.

With respect to $t$ and the associated foliation of space-time, the evolution of the clock, which at $t = 0$, is in an internal state $|s_a(\tau_0)\rangle$, where $\tau_0$ denotes the clock's proper time at $t = 0$, reads

$$e^{-i\Omega_a t\left(1 + \frac{\Phi(R_a - r_M)}{c^2}\right)}|R_a\rangle|s_a(\tau_0)\rangle = |R_a\rangle|s_a(\tau_0 + \tau(R_a - r_M, t))\rangle, \qquad (11)$$

where $\tau(R_a - r_M, t) := t\left(1 + \frac{\Phi(R_a - r_M)}{c^2}\right)$ is the proper time elapsing for the clock at a radial distance $|R_a - r_M|$ from the mass when the elapsed coordinate time is $t$; and for clarity we set $\hbar = 1$.

Before continuing on to the gravitational quantum control, we give an example of an internal Hamiltonian, state, and evolution. Let us take $\Omega_a = E_0|0\rangle\langle 0| + E_1|1\rangle\langle 1|$ and $|s_a(\tau_0 = 0)\rangle = \frac{1}{\sqrt{2}}(|0\rangle + |1\rangle)$, which describe, for example, an atom in an equal superposition of some two electronic energy levels $|0\rangle, |1\rangle$ with energies $E_0$, $E_1$, respectively. Under $H_a$ from Eq. (10) internal state $|s_a(0)\rangle$ from Eq. (11) evolves as

$$\begin{aligned} e^{-i\Omega_a t\left(1 + \frac{\Phi(R_a - r_M)}{c^2}\right)}|s_a(0)\rangle = \ & \frac{1}{\sqrt{2}} e^{-iE_0 t\left(1 + \frac{\Phi(R_a - r_M)}{c^2}\right)}|0\rangle + \frac{1}{\sqrt{2}} e^{-iE_1 t\left(1 + \frac{\Phi(R_a - r_M)}{c^2}\right)}|1\rangle \\ \equiv \ & \frac{1}{\sqrt{2}} e^{-iE_0 \tau(R_a - r_M, t)}|0\rangle + \frac{1}{\sqrt{2}} e^{-iE_1 \tau(R_a - r_M, t)}|1\rangle \end{aligned} \qquad (12)$$

which is simply $|s_a(\tau(R_a - r_M, t))\rangle$.

We now use the above to show how the quantum superposition principle and general relativity lead to the prediction that quantised matter acts as a quantum control of temporal order. To this end, we assume conditions (a)–(c) from the Results section and consider two clocks positioned at $R_A$ and $R_B$, respectively. The Hamiltonian of clock a is thus Eq. (10) and fully analogously for b, $H_b \approx \Omega_b \left(1 + \frac{\Phi(R_b - r_M)}{c^2}\right)$. The clocks are initially synchronised with each other and with a clock of the distant agent so that at $t_0 = 0$ both clocks are at $\tau_0 = 0$. We further consider a target system, for example, a mode of the electromagnetic field, initially in a state $|\psi\rangle^S$, on which an operation $\mathcal{O}_A$ is performed at an event A = ($R_a$, $\tau_a = \tau^*$) and an operation $\mathcal{O}_B$ at an event B = ($R_b$, $\tau_b = \tau^*$), where $\tau_a$, $\tau_b$ refer to the

proper times of the clock A, B, respectively. We effectively represent these operations as $\mathcal{O}_A = \delta(\tau_a - \tau^*, r - R_a)O_A$, where $\delta(\tau_A - \tau^*, r - R_a)$ is a Dirac delta distribution and $O_A$ is an operator (e.g. describing rotation of the polarisation of an electromagnetic field mode by a particular half-wave plate) independent of time and location. The total Hamiltonian reads

$$H_{tot} = H_a + H_b + \mathcal{O}_A + \mathcal{O}_B, \qquad (13)$$

which for simplicity assumes trivial time evolution of the mass and of the target system between the application of the operations. We furthermore consider the following initial (at $t_0 = 0$) state of the mass, clocks and the target system:

$$|\psi(0)\rangle^{MSab} = |R_a\rangle|R_b\rangle|s_a(\tau_0 = 0)\rangle|s_b(\tau_0 = 0)\rangle|\psi\rangle^S \left(|r_L\rangle^M + |r_R\rangle^M\right), \qquad (14)$$

where positions $r_L$, $r_R$ of the mass refer to the configurations in the left and the right panel of Fig. 1, respectively, that is, they realise configurations $K_{A\prec B}$ and $K_{B\prec A}$: for $|r_L\rangle$ the mass is at a distance $r_a = r_L - R_a$ from clock a and at $r_b = r_a - h$ from b, while for $|r_R\rangle$ the relative distances are swapped and the mass is at a distance $r_a - h$ from a and at $r_a$ from b. After coordinate time $t$ such that $\tau(r_a, t) > \tau^*$ (where $\tau^* > \frac{2r_b^2 c}{GM}$, see main text) the state evolves to

$$|\psi(t)\rangle^{MSab} = |R_a\rangle|R_b\rangle \big(|s_a(\tau(r_a, t))\rangle|s_b(\tau(r_a - h, t))\rangle e^{-iO_B} e^{-iO_A}|\psi\rangle|r_L\rangle^M$$
$$+ |s_a(\tau(r_a - h, t))\rangle|s_b(\tau(r_a, t))\rangle e^{-iO_A} e^{-iO_B}|\psi\rangle^S|r_R\rangle^M\big). \qquad (15)$$

The order of applying unitary transformations $U_A = e^{-iO_A}$ and $U_B = e^{-iO_B}$ to the target system is controlled by the position of the mass, which due to time-dilation changes causal relations between events A and B. Swapping the mass distribution: $|r_L\rangle \rightarrow |r_R\rangle$, $|r_R\rangle \rightarrow |r_L\rangle$ and letting the state evolve for another time interval $t$ results in the final state where the clocks become synchronised again

$$|\psi(t)\rangle^{MSab} = |R_a\rangle|R_b\rangle|s_a(\tau_f)\rangle|s_b(\tau_f)\rangle \left(U_B U_A|\psi\rangle^S|r_R\rangle^M + U_A U_B|\psi\rangle^S|r_L\rangle^M\right), \qquad (16)$$

where $\tau_f = \tau(r_a, t) + \tau(r_a - h, t)$. Measuring the mass in a superposition basis $|r_L\rangle \pm |r_R\rangle^M$ prepares the target system in the corresponding superposition state $U_B U_A |\psi\rangle^S \pm U_A U_B|\psi\rangle^S$.

The above example demonstrates that under very conservative assumptions a spatial superposition of a mass generates a quantum-controlled application of unitary operations. More fundamentally, this effect stems from the superposition of different causal structures associated with the superposed states of the mass.

**Proof of Bell's theorem for temporal order.** Bell's theorem in general asserts that, under certain assumptions, the correlations between the outcomes of independent measurements on two subsystems must satisfy a class of inequalities. The two measuring parties are referred to as Alice and Bob. In every experimental run, each of them measures one of two properties of the subsystem they receive. For each of the properties, one of two outcomes is obtained, for convenience chosen to be ±1. Bell's inequalities follow from the conjunction of the following assumptions: (1) measurement results are determined by properties that exist prior to and independent of the experiment (hidden variables); (2) results obtained at one location are independent of any measurements or actions performed at space-like separation (locality); (3) any process that leads to the choice of which measurement will be carried out is independent from other processes in the experiment (free choice). The outcomes of Alice $A(i, \lambda)$ and Bob $B(i, \lambda)$ thus only depend on their own choice of setting, index $i$, and on the property of the system, variable $\lambda$. The correlation between outcomes $A(i, \lambda)$ and $B(i, \lambda)$ for the measurement choices $i$, $j$ is described by $E(A_i, B_j) = \int d\lambda P(\lambda)A(i, \lambda)B(j, \lambda)$, where $P(\lambda)$ is the probability distribution over the properties of the systems. It is straightforward to check that one possible inequality satisfied by the correlations $E(A_i, B_j)$ is the so-called Clauser–Horne–Shimony–Holt inequality: $|E(A_1, B_1) + E(A_1, B_2) + E(A_2, B_1) - E(A_2, B_2)| \leq 2$. Crucially, quantum theory allows for the left-hand side of this inequality to reach a value >2, and experimental verification of this (and other inequalities) have confirmed such a violation[3–6]. The significance of the violations of Bell's inequalities is in showing that neither nature nor quantum mechanics obey all three assumptions mentioned above.

The assumption of classical order is sufficient to derive Causal Inequalities[16,60]: tasks that, without any further assumptions, cannot be performed on a classical causal structure. However, it is not enough to violate causal inequalities using quantum control of order[45,61], this is why additional assumptions were required in the present context. It is an open question whether a gravitational implementation of a scenario that does allow for a violation of causal inequalities is possible.

The theorem we have formulated is theory independent, but not fully device-independent, as it requires the notions of a physical state and a physical transformation (in addition to the measured probability distributions), which we introduce below and then proceed to the proof. Discussion of the present work in the context of the theory-dependent framework of causally non-separable quantum processes[16,45,61] and the fully theory- and device-independent approach of causal inequalities[16,60] is presented in Supplementary Note 1.

We consider a sufficiently broad framework to describe physical systems that can undergo transformations and measurements, similar to generalised probabilistic theories[62–64]. This framework is more general than quantum or classical theory and we thus need to define key notions required in the proof. In this framework, a state $\omega$ is a complete specification of the probabilities $P(o|i, \omega)$ for observing outcome $o$ given that a measurement with setting $i$ is performed on the system. We are interested in situations where a system can be split up

in subsystems, say $S_1$ and $S_2$, with space-like separated agents performing independent operations on $S_1$ and $S_2$. We say $\omega$ is a product state, and write $\omega = \omega_1 \otimes \omega_2$, if probabilities for local measurements factorise as $P(o_1, o_2|i_1, i_2, \omega) = P(o_1|i_1, \omega_1)P(o_2|i_2, \omega_2)$. If state $\omega_1^f$ is prepared for system $S_1$ and state $\omega_2^f$ is prepared for system $S_2$, according to a probability distribution $P(f)$ for some variable $f$, we write $\omega = \int df P(f)\omega_1^f \otimes \omega_2^f$ and say the state is separable. Probabilities are then given by the corresponding mixture: $P(o_1, o_2|i_1, i_2, \omega) = \int df P(o_1|i_1, \omega_1^f)P(o_2|i_2, \omega_2^f)P(f)$. Note that for such a decomposition Bell inequalities cannot be violated[1,65].

A physical transformation of the system is represented by a function $\omega \mapsto T(\omega)$. To make our arguments precise we need a notion of local transformations, namely, realised at the time and location defined by a local clock. If $S_1$ is the subsystem on which a local transformation $T_1$ acts, and $S_2$ labels the DOFs space-like separated from $T_1$, then, by definition, $T_1$ transforms product states as $\omega_1 \otimes \omega_2 \mapsto T_1(\omega_1) \otimes \omega_2$ and separable states by convex extension. How local operations act on general, non-separable states can depend on the particular physical theory; however, action on separable states will suffice for our purposes. We further need to define how the local transformations combine. This depends on their relative spatio-temporal locations: if transformations $T_1$, $T_2$ are space-like separated they combine as $(T_1 \otimes T_2)(\omega_1 \otimes \omega_2) = T_1(\omega_1) \otimes T_2(\omega_2)$, which follows from the definition above; if $T_1$ is in the future of $T_2$, we define their combination as $T_1 \circ T_2(\omega) = T_1(T_2(\omega))$. (For simplicity, we omit possible additional transformations taking place between the specified events, as they are of no consequence for our argument.)

*Proof* Assumption (1) says that there is a random variable $f$ determining the local states $\omega_1^f$, $\omega_2^f$ of systems $S_1$, $S_2$, respectively. Assumption (3) says there is a random variable $\lambda$ that determines the order of events. In general, the two variables can be correlated by some joint probability distribution $P(\lambda, f)$. By assumption (4), events labelled $A_1$, $B_1$ are space-like separated from events $A_2$, $B_2$ and the order between events within each set $(A_j, B_j)$, $j = 1, 2$ can be defined by a permutation $\sigma_j$. Most generally, there is a probability $P(\sigma_j|\lambda)$ that the permutation $\sigma_j$ is realised for a given $\lambda$. By assumption (2), for each given order the system undergoes a transformation $T^{\sigma_1} \otimes T^{\sigma_2}$, where $T^{\sigma_1}$ is the transformation obtained by composing $T_{A_1}$ and $T_{B_1}$ in the order corresponding to the permutation $\sigma_1$ and similarly for $T^{\sigma_2}$. (For example, if $\sigma_1$ corresponds to the order $A_1 \prec B_1$, then $T^{\sigma_1} = T_{B_1} \circ T_{A_1}$.) Furthermore, at event D an outcome $z$ is obtained with a probability $P(z|\lambda, f, \sigma_1, \sigma_2)$. Finally, using assumption (1), we write the probabilities for all outcomes as

$$P(o_1, o_2, z|i_1, i_2, \mathbb{T}, \omega) =$$
$$\sum_{\sigma_1 \sigma_2} \int d\lambda\, df P(o_1|i_1, T^{\sigma_1}(\omega_1^f))P(o_2|i_2, T^{\sigma_2}(\omega_2^f))P(\sigma_1|\lambda)P(\sigma_2|\lambda)P(z|\lambda, f, \sigma_1, \sigma_2)P(\lambda, f). \qquad (17)$$

A simple Bayesian inversion $P(\sigma_1|\lambda)P(\sigma_2|\lambda)P(z|\lambda, f, \sigma_1, \sigma_2)P(\lambda, f) = P(\lambda, f, \sigma_1, \sigma_2|z)P(z)$, where we used $P(\sigma_j|\lambda) = P(\sigma_j|\lambda, f)$, gives the desired probabilities

$$P(o_1, o_2|i_1, i_2, z, \mathbb{T}, \omega) = \sum_{\sigma_1 \sigma_2} \int d\lambda\, df P(o_1|i_1, T^{\sigma_1}(\omega_1^f))P(o_2|i_2, T^{\sigma_2}(\omega_2^f))P(\lambda, f, \sigma_1, \sigma_2|z)$$
$$= \int d\tilde{f} P(o_1|i_1, T^{\sigma_1})P(o_2|i_2, T^{\sigma_2})P(\tilde{f}|z), \qquad (18)$$

where $\tilde{f}$ is a short-hand for the variables $\lambda$, $f$, $\sigma_1$, and $\sigma_2$. The above probability distribution satisfies the hypothesis of Bell's theorem and thus cannot violate any Bell inequality.

**Exemplary scenario realising Bell test for temporal order of events.** The protocol allowing for the violation of Bell's inequalities for temporal order exploits correlations between the clocks of the agents $a_1$, $b_1$ and the agents $a_2$, $b_2$, created due to time dilation induced by the mass. It should be noted that the protocol allows maximal violation of the Bell inequality if the joint state of the systems $S_1$ and $S_2$ is pure (and maximally entangled) when the Bell measurements are realised. Thus, for a maximal violation, the clocks need to decorrelate from the mass after the application of the unitaries. Below we sketch a scenario that can achieve this.

The space-time arrangement of the mass and the agents in this example is presented in Fig. 4. It can be realised in one spatial dimension: agents acting on the system $S_1$ are located at distance $h$ from each other, and the mass is placed at distance $r$ (configuration $K_{B\prec A}$) or $r + L$ (configuration $K_{A\prec B}$) from agent $a_1$. Agents acting on system $S_2$ are placed symmetrically on the opposite side of the mass, such that the mass is at a distance $r + L$ from $a_2$ in configuration $K_{B\prec A}$ and $r$ in configuration $K_{A\prec B}$. Here, events $A_j$ are defined by the local time $\tau_a$ that differs from the local time $\tau_b$ defining $B_j$, $j = 1, 2$. In such a case, even though the mass is always closer to $a_j$ than to $b_j$, the two mass configurations can lead to different event orders—as they induce different relative time dilations. (Equivalently, one can introduce an initial offset in the synchronisation of the clocks.) Note that the time orders between the two groups are here "anti-correlated": $A_1 \prec B_1$ and $B_2 \prec A_2$ for $K_{A\prec B}$, and vice versa for $K_{B\prec A}$. Since otherwise the scenario is the same for $S_1$ and $S_2$, we focus on the operations performed on $S_1$. The key observation is that swapping the mass distribution, as depicted in Fig. 4, will eventually disentangle the clocks from the mass, and since the clocks must be suitably time dilated when the operations are performed, the operations must not take place in the future light cone of the swapped mass state.

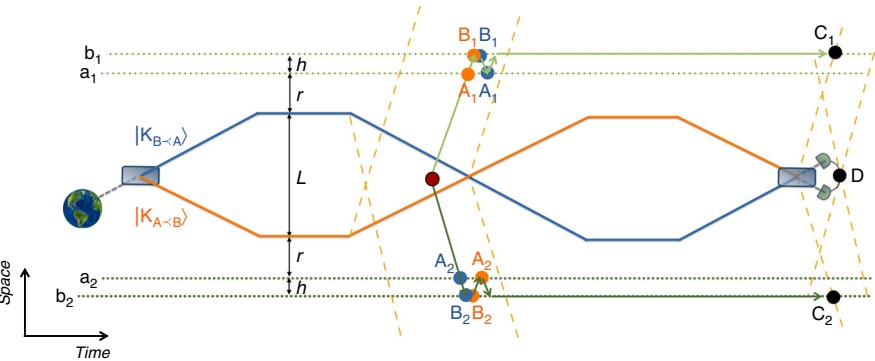

**Fig. 4** Space-time diagram of a protocol for disentangling the clocks from the mass. In configuration $K_{A \prec B}$ the mass is at a distance $r + L$ from $a_1$, and at $r + L + h$ from $b_1$. In $K_{B \prec A}$—it is at $r$ from $a_1$ and at $r + h$ from $b_1$. The opposite holds for $a_2$, $b_2$. The initial mass superposition is swapped (after sufficient time to prepare the clocks in the correlated state) so that they finally show the same time. At the local time $\tau_a$ of $a_1$ (at event $A_1$) the agent applies $U_{A_1}$ on $S_1$. At the local time $\tau_b$ of $b_1$ the agent applies $U_{B_1}$ on $S_1$. For the mass configuration $K_{A \prec B}$ $A_1$ is before $B_1$ (orange-coloured events), while for $K_{B \prec A}$ event $B_1$ is before $A_1$ (blue-coloured events). The opposite order holds for events $A_2$, $B_2$ occurring on the opposite side of the mass, where agents $a_2$, $b_2$ act on $S_2$. Unitary operations should be applied in the future light cone of the event where the clocks get correlated and outside the future light cone of the event when the mass amplitudes are swapped, Bell measurements (at $C_1$, $C_2$) should be made when the clocks become disentangled (at future light-like events to when the mass amplitudes are brought together), and the measurement at event D should be space-like to $C_1$, $C_2$; dashed yellow lines represent the relevant light cones

The proper time $\tau_a$ that has to elapse for the clock of $a_1$ such that the order of events is $A_1 \prec B_1$ for $|K_{A \prec B}\rangle$ and $B_1 \prec A_1$ for $|K_{B \prec A}\rangle$ for the present case reads

$$\tau_a = \sqrt{-g_{00}(r)} \frac{T_c(r,h) + T_c(r+L,h)\sqrt{\frac{g_{00}(r+L+h)}{g_{00}(r+h)}}}{1 - \sqrt{\frac{g_{00}(r)g_{00}(r+L+h)}{g_{00}(r+h)g_{00}(r+L)}}}, \quad (19)$$

where $T_c(r, L/2)$ is the coordinate travel time of light between radial distances $r$ and $r + L/2$ from the mass. The coordinate time corresponding to $\tau_a$ is $T_a = \tau_a/\sqrt{-g_{00}(r)}$. The proper time of event $B_1$ is then defined as:

$$\tau_b = \sqrt{-g_{00}(r+L+h)} \left( \frac{\tau_a}{\sqrt{-g_{00}(r+L)}} + T_c(r+L,h) \right). \quad (20)$$

It can directly be checked that when the mass is placed in configuration $K_{A \prec B}$—at a distance $r + L$ from $a_1$—the event $A_1$ defined by local clock of $a_1$ showing proper time $\tau_a$ from Eq. (18) is in the past light cone of event $B_1$, which is defined by the local clock of $b_1$ showing proper time $\tau_b$ from Eq. (19). When the mass is placed in configuration $K_{B \prec A}$, event $B_1$ ends up in the past of the event $A_1$. The coordinate time required for the application of the operations can be estimated as twice the travel time of light between the agents, $T_o = 2T_c(r + L/2, h)$.

The world lines of the mass can be arranged such that: (a) the mass is moving slow so that the two amplitudes of the mass are swapped in a time interval longer than $T_o$; (b) during the application of the operations the distance of each agent to the mass is approximately the same for both mass configurations (as in Fig. 4). The first guarantees that there is enough time to apply the operations after the clocks get correlated, the second—that the slow-down of light in curved space-time, the Shapiro delay[66,67], can be neglected.

The coordinate-time duration of the entire protocol can be estimated as $T_p = 2T_a + 4L/2c$, where $L/2c$ is the minimal time required to put the mass in superposition of amplitudes separated by the distance $L/2$. Taking as an example $M \sim 0.1 \, \mu g$, $L = h \sim 0.1 \, \mu m$, $r \sim 1$ fm, the protocol in Fig. 4 takes $T_p \sim 10$ h. Furthermore, we note that a quantum treatment of the local clocks is central to our protocol since the application of the operations on the target systems is conditioned on the states of the clocks. The time-energy uncertainty[68,69] thus poses a limitation to a single-shot precision with which space-time events can be defined with physical clocks. The optimal clock state in this context—evolving the fastest—is a balanced superposition of energy eigenstates; for an energy gap $h \cdot 2\pi\nu_c$, where $\nu_c$ is the clock frequency, the smallest time that can be resolved by a single quantum system is the so-called orthogonalisation time[70–72] $t_\perp = 1/2\nu_c$. For the values of parameters quoted above, the coordinate-time difference between the superposed locations of the events $A_i$, $i = 1, 2$ is $\sim 10^{-15}$ s, and we thus need a system with frequency $\nu_c \geq 10^{15}$ Hz such as a clock based on optical transitions in ytterbium[73] or mercury[74], which both give $t_\perp \sim 10^{-16}$ s. While this ideal limit is not reached with practical systems, the resolution of current atomic clocks based on such atoms far exceeds this theoretical bound due to averaging over many atoms, with $2.5 \times 10^{-19}$ uncertainty of the clock frequency recently demonstrated in ref. [75]. We further note that by using $n$ entangled atoms, the orthogonalisation time of the entire system becomes $t_\perp/n$ and can thus be even a few orders of magnitude smaller[76] than required. Finally, such atoms have masses $\sim 10^{-25}$ kg and their back action on the metric produced by $M \sim 10^{-7}$ kg would thus be negligible. Since the mass difference between the atom in the two involved energy levels is $2\pi\hbar\nu_c/c^2 \sim 10^{-35}$ kg

also quantum effects from the clocks' mutual gravitational interactions[58] can be neglected.

We conclude that it is in principle possible to achieve the required entanglement of orders, swap the mass distribution so as to finally disentangle the clocks from the mass, and satisfy the locality conditions on the events. Although a direct experiment in such a regime is not practical, the above example surprisingly shows that the regime where entangled temporal order arises is in no way related to the Planck scale. It is usually assumed that the Planckscale marks the regime where quantum gravity effects become relevant (first discussed in this context by Bronstein[77]), but this is not the case for the superposition of temporal order. In terms of a potential experiment, one could also take a different (theory-dependent) approach and explore possible witnesses of entangled temporal order[61], in analogy to witnesses of entanglement in quantum-information theory[78]. A witness would probe the quantum nature of temporal order indirectly and under further assumptions, but in a relaxed parameter range. Such an approach may lead to more feasible experiments, which will be explored in a future study.

A spatial superposition state of a mass such as used in our protocol is postulated to decohere in various gravity-inspired collapse models[9–13] (which thus violate assumption (c) in the first section). However, even if endorsed, these models do not immediately preclude realisation of our protocol: the decoherence time scale in those models is the Diosi–Penrose time[10,11] $T_{DP} = \frac{2\delta^3\hbar}{G(ML)^2}$, where $\delta$ is a free parameter. For every value of $\delta$ one can find the mass and the relevant distances $(M, r, L, h)$ so that the duration of our entire protocol is shorter than $T_{DP}$. For example, following the recent ref. [79] and taking $\delta = 10^{-7}$ m, for $r = 10^{10}R_{Sch}$, $L = 5r$, $h = r$ and $M = 1$ g, where $R_{Sch} \approx 10^{-30}$ m, the protocol from Fig. 4 takes $T_p \approx 7 \times 10^{-18}$ s, while $T_{DP} \approx 0.5$ s. Taking instead the originally proposed value $\delta = 10^{-15}$ m[10], the desired regime is achieved, for example, for $M = 10^{-7}$ kg, $r = 10^7 R_{Sch}$, $L = 5 \times 10^5 r$, $h = 10^5 r$; with $T_p \sim 10^{-23}$ s and $T_{DP} \sim 10^{-13}$ s. Thus, the above models in principle still allow for events with entangled temporal order, and do not enforce the classicality of the causal structure of space time.

## Data availability
The data that support the plots within this paper and other findings of this study are available from the corresponding author upon reasonable request.

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

## Acknowledgements

We thank G. Chiribella, G. Milburn, H. Wiseman and M. Vojinovic for feedback. M.Z. and F.C. acknowledge support through the Australian Research Council (ARC) Centre of Excellence for Engineered Quantum Systems (CE 110001013), Discovery Early Career Researcher Awards DE180101443, DE170100712 and the Templeton World Charity Foundation (TWCF 0064/AB38). I.P. acknowledges support of the NSF through a grant to ITAMP and the Branco Weiss Fellowship—Society in Science, administered by the ETH Zürich. Č.B. acknowledges the support of the Austrian Science Fund (FWF) through the Doctoral Programme CoQuS, the project I-2526-N27 and I-2906, the research platform TURIS and the ÖAW Innovationsfond "Quantum Regime of Gravitational Source Masses". This publication was made possible through the support of a grant from the John Templeton Foundation and from the Foundational Questions Institute (FQXi) Fund. The opinions expressed in this publication are those of the authors and do not necessarily reflect the views of the John Templeton Foundation. F.C. and M.Z. acknowledge the traditional owners of the land on which the University of Queensland is situated, the Turrbal and Jagera people.

## Author contributions

M.Z., F.C., I.P and Č.B. contributed to all aspects of the research, with the leading input from M.Z.

## Additional information

**Competing interests:** The authors declare no competing interests.

