## [Peer Review File · Nature Communications]

Reviewers' comments:

Reviewer #5 (Remarks to the Author):

This is an interesting article that addresses a foundational topic, of interest to the quantum gravity and quantum gravitational communities. It addresses the crucial issue of identifying the properties of the gravitational field that is generated by quantum matter, at the weak gravity non-relativistic limit that is accessible to experiments. The authors focus on the behavior of the causal/temporal structure associated to the spacetime geometry, and they prove that under specific assumptions, it is possible to prove a version of Bell's theorem for the causal order of two specific events. The effect is too small to be measurable, but the whole issue is important, even if one thinks of it as a thought experiment for an important conceptual issues, rather than as a concrete experimental proposal.

The authors present a good argument. Its essence is the following. If the gravitational field that is generated by a quantum source preserves some quantum features, then it will induce this quantum behavior to other quantum systems; in particular, it may cause quantum behavior to the causal order of specific events that can be encoded into Bell inequalities.

However, I have two reservations about this result. First, I am not convinced that the Bell inequality derived in the paper is specifically gravitational. The essential requirement for deriving equation (4) is that a quantum system can interact with a quantum clock (or with any system that involves the measurement of a temporal variable) and, as a result, change its reading (or the probability distribution of readings). Gravity is not essential to the indefiniteness of the causal order of two events. If this is the case, much of the reference to gravity in this paper is misleading: the indefinite causal ordering discussed here is not directly relevant to the spacetime causal structure of General Relativity.

Second, I understand that the authors treat the clocks as quantum systems (it was difficult to find a precise statement in the text, but they make this point in a reply to a referee). This means that clocks are subject to uncertainties and they cannot reflect the spacetime causal structure with full accuracy. There is an intrinsic indefiniteness to the causal order due to the quantumness of the clock, even in absence of an external mass.

This indefiniteness is related to the clock's time uncertainty, and hence to its energy uncertainty by the Mandelstam-Tamm inequality. The latter is bounded above by the clock's mass. There is a good a priori possibility that the intrinsic quantum uncertainties may be stronger than the uncertainty in the causal order generated by the external mass, since the latter is so weak. The authors must demonstrate that there is at least one regime in which this is not the case. Otherwise, quantum control of the temporal order in a gravitational set-up may turn out to be impossible.

The authors should address these points before the article can be considered for publication.

Reviewer #6 (Remarks to the Author):

This article provides an explicit analysis of the quantum causal structure arising from a quantum superposition of gravitating masses. It is shown that the resulting quantum behaviour of spacetime may affect the behaviour of time-like separated events whose temporal order can become superposed or entangled. A Bell-type theorem for temporal order is formulated and it is shown that this entanglement allows to accomplish a task -the violation of a Bell inequality- that

would not be accomplished if the time-order between events were determined by classical local variables.

Even though quantum gravity specialists are well aware of the quantum nature of the causal structure, a detailed analysis of a concrete physical situation based on a small set of plausible assumptions about the low energy behaviour of general relativity and quantum mechanics is new and useful for testing the existing approaches to the quantization of gravity. It is important to remark that the paper should not be taken as a proposal of a realistic experiment about a quantum gravitational situation. However the paper convincingly shows that observable quantum gravitational effects that require extreme conditions but do not involve Planck scale phenomena are possible.

The previous rounds of review at [redacted] have led to important clarifications and significant improvements in the presentation of the manuscript.

Concerning the issues raised by referee #3, I will discuss what I consider the two main objections:

a) the authors consider a particular time foliation that could in principle depend on the possible positions of the mass; b) the authors need to assume that the agents move along fixed world-lines, independently of the possible positions of the mass, which seems difficult to implement. In what concerns the first objection, I consider that the discussion included by the authors in section I of the article and Methods section A, gives a satisfactory answer to this concern. In fact the use of a foliation of space-time and a common coordinate system is compatible with having many metrics and consequently many causal structures. Only observable operators are defined on the physical Hilbert space of a system. The metric operator –and the causal structure– depend on these observables and on the particular choice of coordinates that one has decided to use.

In what concerns the second objection, the authors offer two different implementations for keeping the world-lines fixed regardless of the gravitational fields induced by the fluctuating mass position. The first one that appears in section I, consists in embedding the agents in a “tight enough trapping potential”. The second implementation presented in the supplementary material uses a superposition of massive shells in order to ensure that agents remain inertial. At least the first implementation seems feasible in principle, for the parameter regimes detailed in Methods, section C. My conclusion with respect to the concerns of referee #3 is therefore that they have been satisfactorily addressed.

The issues posed by referees #4 and #1 refer to the unrealistic character of the experiment. The scenario discussed in Methods C, requires superposed positions of a mass with densities several orders of magnitude higher than neutron stars and agents able to act at scales smaller than the nanometres, which are completely out of reach of the actual technologies, fact that the authors admit. They are only claiming that the violation of the Bell inequality is in principle observable and that there is not any fundamental reason that forbids “a physical manifestation of non-classical temporal order.”

Let me now summarize the reasons that lead me to recommend the publication of this paper: i) Based on a simple and explicitly established set of assumptions basically related with the linearity of quantum theory in a post-Newtonian gravitational regime the authors show that the temporal order between time-like separated events may become superseded and that entanglement of temporal orders is possible. These conclusions seem to be robust across to different quantum gravity frameworks. ii) The authors propose a thought experiment showing that this kind of entanglement allows accomplishing a task, the violation of a Bell inequality. It is an, in principle, observable effect that requires extreme conditions for a realistic experimental verification.

However these conditions are less extreme than the Planck scale. The thought experiment does not require, as most of the proposals on quantum gravity phenomenology do, exotic conditions: like violations of quantum mechanics or Lorentz invariance, black holes produced in colliders, among many others. See for instance {Jacobson et al. Springer Proc.Phys. 98 (2005) 83-98}.

iii) Another important and original contribution of the paper is the study of the physical implications of quantum entangled causal structures. The proof that entangled causal structures suffices for a maximal violation of the Bell’s inequality is an explicit demonstration that one of the main features of quantum mechanics may be extended to cases where a classical notion of temporal order does not exist. Assumptions like the one made by Penrose about the state reduction of systems that could show superpositions of causal structures are not required, and there exists a consistent phenomenology associated with quantum causal structures.

I recommend the publication of the paper with the suggestion of introducing an explicit comment about the fact that the “work proposes a thought experiment rather than a ready-to-implement set-up” as the authors recognize repeatedly in the answer to the referees.

We thank the referees for a careful reading of our manuscript and for the useful suggestions. We address the raised issues in our replies below. We have revised our manuscript accordingly, with the changes marked in red.

Reviewer #5 (Remarks to the Author):

However, I have two reservations about this result. First, I am not convinced that the Bell inequality derived in the paper is specifically gravitational. The essential requirement for deriving equation (4) is that a quantum system can interact with a quantum clock (or with any system that involves the measurement of a temporal variable) and, as a result, change its reading (or the probability distribution of readings). Gravity is not essential to the indefiniteness of the causal order of two events. If this is the case, much of the reference to gravity in this paper is misleading: the indefinite causal ordering discussed here is not directly relevant to the spacetime causal structure of General Relativity.

Reply:

The crucial aspect of gravity is that its effect is universal. The heart of the protocol to produce eq (4) is time dilation of the local clocks. It is indeed true that, given a specific physical clock, other forces can affect its ticking rate, analogously to gravitational time dilation. For example, an electric field can shift atomic energy levels and thus “time dilate” a clock based on atomic transitions, a magnetic field can change the rate of spin precession and so “time dilate” a clock based on spin precession. In this way, the scenario we are describing can be mimicked by a specific non-gravitational interaction for a specific system. However, no mechanism other than gravity can cause a universal change in clocks, regardless of their composition: gravity will equally time dilate spin-based clocks, atomic clocks, clocks based on nuclear or chemical processes, etc. Therefore, only gravity changes the space-time causal structure, no other interactions (that we know of) have such an effect.

In our scenario, the clocks used in the measurements can be arbitrary and will lead to the exact same outcome, eq (4), only if gravity is employed as a control system. Thus, while the effect can be “simulated” using a specific clock mechanism and suitably chosen fields, using other clocks in such scenarios would not result in eq (4) and reveal that the causal structure itself is not affected. (In more technical terms, local events can have “indefinite order” with respect to specific clocks manipulated using suitably chosen non-gravitational interactions but in such a case one always finds that events defined with respect to other clocks, and in particular with respect to the space-time background, do have a classically well-defined order.) Only the gravitational version is actually changing the causal structure and will result in the outcome desired in our manuscript. In other words, only the gravitational potential is interpreted in terms of a space-time metric precisely because of its universal effect on all other systems and interactions and this universality is crucial to our work.

We have added a paragraph clarifying this aspect under eq (4), on page 7 in the revised manuscript.

Second, I understand that the authors treat the clocks as quantum systems (it was difficult to find a precise statement in the text, but they make this point in a reply to a referee). This means that clocks are subject to uncertainties and they cannot reflect the spacetime causal structure with full accuracy. There is an intrinsic indefiniteness to the causal order due to the quantumness of the clock, even in absence of an external mass. This indefiniteness is related to the clock's time uncertainty, and hence to its energy uncertainty by the Mandelstam-Tamm inequality. The latter is bounded above by the clock's mass. There is a good a priori possibility that the intrinsic quantum uncertainties may be stronger than the uncertainty in the causal order generated by the external mass, since the latter is so weak. The authors must demonstrate that there is at least one regime in which this is not the case. Otherwise, quantum control of the temporal order in a gravitational set-up may turn out to be impossible.

Reply:

We thank the referee for this helpful and relevant comment; indeed the quantum uncertainties of the clocks themselves were not discussed in our work. We have rectified this by adding the relevant calculations along the lines suggested by the referee (quantifying the clock's precision in terms of time-energy uncertainty). The results are summarized in a new paragraph in Methods section, page 19. We show that given the specific quantitative example already present therein, the required clock's precision would be of the order of 10^{-15} seconds, which is achievable with a clock based e.g. on electronic energy transitions in an atom. Thus, the intrinsic clock uncertainty can be smaller than the effect we describe in our manuscript and time-energy uncertainty does not fundamentally prevent a realization of the protocol.

We would like to further highlight that with such an implementation, the clocks can be effectively treated as passive "probe systems" as they are many orders of magnitude lighter than the mass used to control the time order. Their backaction on the external space-time as well as their mutual gravitational interactions can thus be negligible. This is, however, by itself an interesting research topic and some of its aspect were discussed, e.g., by Castro-Ruiz et al. *PNAS* **114**, E2303–E2309 (2017), ref [74] in the revised draft.

Reviewer #6 (Remarks to the Author):

I recommend the publication of the paper with the suggestion of introducing an explicit comment about the fact that the “work proposes a thought experiment rather than a ready-to-implement set-up” as the authors recognize repeatedly in the answer to the referees.

Reply:

We thank the referee for the comprehensive discussion of our work and for the specific suggestion. We are convinced that it will indeed help us clarify the scope of our work. Following the recommendation we have reworded the manuscript in several places (among other – in the abstract and on page 2 of the introduction of the revised manuscript) to clarify the status of our scenario as a thought experiment rather than an experimental proposal.

REVIEWERS' COMMENTS:

Reviewer #5 (Remarks to the Author):

In my previous communication, I had asked the authors for a clarification of two points. The reply to the second query (the relevance of the time-energy uncertainty relation) was fully satisfactory. I am not fully convinced about the reply to the first one. My point was that the phenomena of indefinite ordering of events here may not be specifically gravitational, hence, they must first be understood in their general context, before the special case of gravity is considered. I think that it is important to clarify that the indefinite causal order of events is NOT a specifically gravitational phenomenon, but a rather common phenomenon for quantum systems that evolve in time. The universality of gravity may be a differentiating factor, as the authors claim, but this would require further analysis, going well beyond the scope of this paper.

My disagreement on this point does not change my overall positive appraisal of the paper, and for this reason, I recommend its publication.

We thank the referee for the positive recommendation and for reviewing our arguments and replies.

Reviewer #5 (Remarks to the Author):

In my previous communication, I had asked the authors for a clarification of two points. The reply to the second query (the relevance of the time-energy uncertainty relation) was fully satisfactory. I am not fully convinced about the reply to the first one. My point was that the phenomena of indefinite ordering of events here may not be specifically gravitational, hence, they must first be understood in their general context, before the special case of gravity is considered. I think that it is important to clarify that the indefinite causal order of events is NOT a specifically gravitational phenomenon, but a rather common phenomenon for quantum systems that evolve in time. The universality of gravity may be a differentiating factor, as the authors claim, but this would require further analysis, going well beyond the scope of this paper.

My disagreement on this point does not change my overall positive appraisal of the paper, and for this reason, I recommend its publication.

Reply:

We were happy to read that the Reviewer found our reply to their first question satisfactory.

We acknowledge and share the Reviewer's opinion that in order to answer in full their second question – regarding the relation between non-gravitational and gravitational indefinite order of events – a separate study is required. While we discuss this question in the section Discussion of the main manuscript and in Supplementary Note 4, we have now also added an additional sentence on page 12 (penultimate paragraph) of the main manuscript, stating that:

“The full extent of the relation between gravitational and nongravitational realisations of quantum causal structures merits an in-depth study that goes beyond the scope of the present work and will be explored elsewhere.”